# The Effects of Strength and Conditioning in Physical Education on Athletic Motor Skill Competencies and Psychological Attributes of Secondary School Children: A Pilot Study

**DOI:** 10.3390/sports8100138

**Published:** 2020-10-17

**Authors:** Ben J. Pullen, Jon L. Oliver, Rhodri S. Lloyd, Camilla J. Knight

**Affiliations:** 1Youth Physical Development Centre, Cardiff Metropolitan University, Cyncoed Campus, Cardiff CF23 6XB, Wales, UK; B.Pullen@outlook.cardiffmet.ac.uk (B.J.P.); rlloyd@cardiffmet.ac.uk (R.S.L.); 2Welsh Institute of Performance Science, Sport Wales, Sophia Gardens, Cardiff CF11 9SW, Wales, UK; c.j.knight@swansea.ac.uk; 3Sports Performance Research Institute, New Zealand (SPRINZ), AUT University, Auckland 0632, New Zealand; 4Centre for Sport Science and Human Performance, Waikato Institute of Technology, Hamilton 3200, New Zealand; 5School of Sport and Exercise Sciences, Swansea University, Fabian Way, Swansea SA1 8EN, Wales, UK

**Keywords:** physical literacy, physical training, children, intervention, physical activity

## Abstract

Leading global physical activity guidelines advocate that young children need to engage in activities that strengthen musculoskeletal tissues and improve movement skill competency. The purpose of this study was to examine the effects of delivering strength and conditioning as part of the physical education curriculum on athletic motor skill competencies (AMSC), physical performance, and psychosocial factors. Forty-six school children aged 11–14 were included in the study, and sub-divided firstly by sex and then into intervention and control groups. Intervention groups received nine lessons of strength and conditioning based activities over a six-week period, while the control groups continued with traditional physical education curricula. The resistance training skills battery (RTSB) and tuck jump assessment (TJA) assessed AMSC. Standing long jump distance assessed lower limb strength, and online surveys examined motivation, physical self-efficacy and self-esteem. Male and female intervention groups significantly improved RTSB (*p* > 0.05) whereas no changes were observed in the control groups. No changes were observed in the intervention groups TJA and only trivial and small non-significant changes in standing long jump performance. Significant increases in motivation of the male intervention group occurred. Strength and conditioning integrated in physical education can improve AMSC in short-term interventions.

## 1. Introduction

Physical activity is known to be effective at improving markers of physical health, fitness and mental health in youths [1,2]. Recommendations by the World Health Organization suggest children should be exposed to activities that develop musculoskeletal tissue and improve movement control at least three times weekly [3]. Meanwhile, in the UK, the National Health Service recommends children aged 5–18 years should aim to develop muscular strength and movement skills [4]. Failure to develop motor skills may lead to a decline in physical activity, as positive associations exist between motor skills and physical activity in children and adolescents [5]. Adolescence, the period in which bodily functions structurally and functionally become adult—between ages eight to 19 years in girls and 10 to 22 years in boys [6]—is marked with large declines in physical activity, particularly after the age of 12 years [7].

In the quest to combat youth physical inactivity, physical literacy has emerged as a central tenet [8,9]. However, physical literacy’s use in many domains has led to some disparity in its meaning. A review by Edwards et al. [10] provides clarity and insight into the definition of physical literacy with two paradigms emerging. The first paradigm based on Whitehead’s research recognises physical literacy as the motivation, confidence and physical competence to engage in physical activity, with individuals being able to skillfully interact with familiar and unfamiliar environments [11,12]. The other paradigm, the long-term athletic development approach [13], focuses on the physical components of physical literacy [10]. Significantly, physical competency has been identified across multiple domains as integral to developing physical literacy [10].

Consistently emerging in the literature as the foundations to physical competency are the fundamental movement skills (FMS) (locomotion, object control and balancing skills), which form the building blocks of more complex movement skills [5,10,14,15]. Youth FMS competency demonstrates a strong positive association to physical activity levels [5,15], yet there is a need to extend the current FMS spectrum [16]. Thus, within the field of strength and conditioning the athletic motor skill competencies (AMSC) have emerged, extending from the previous FMS [17].

Developing AMSC provide youths the foundational movements needed to safely participate in physical activity and improve physical literacy. The demands of sports and physical activity require youths to be able to produce and absorb force in multiple contexts. For instance, the process of jumping and landing from a tree exposes children to high impact forces in the lower limbs. In sports, decelerating, reaccelerating, changing direction, jumping and landing in locomotor activities requires the ability to rapidly absorb and produce force both unilaterally and bilaterally. Ergo, the AMSC provide the foundational movements such as being able to work bilaterally and unilaterally, in both the upper and lower body, to jump and land with sound mechanics, while controlling the core. Improving the strength and movement skills of children and adolescents is an important public health outcome. Furthermore, myths and safety concerns regarding youth populations engaging in resistance training have been dispelled [18,19].

The AMSC are underpinned by the need to produce and absorb force, requiring youths to develop adequate strength to support movement control [20]. For example, a recent study showed that secondary school children with low levels of strength were eight times more likely to have low levels of movement competency compared to pupils with high levels of strength [21]. Therefore, it is conceivable that introducing strength and conditioning activities into the secondary school curriculum can help to improve AMSC in youth populations. A recent meta-analysis has confirmed that resistance training improves the FMS of youth, albeit with the authors identifying that the included studies only reported product orientated measures as opposed to process-orientated measures [22]. Product-orientated assessments examine outcome measures dependent on strength and speed constructs (distance jumped, time/distance), while in contrast, process-orientated measures focus on the quality and technique of the movement skill using movement criteria as a reference [5]. Process-orientated assessments of AMSC reduce maturation biases in performance, particularly the sex driven differences such as increased force production capacity in males and reduced neuromuscular control in females [23,24].

To satisfy the need to develop AMSC in youths, due to its inclusivity, school physical education should be considered. Physical education aims to create life-long participation in physical activity and is a powerful tool to develop physical competency in youths [25,26]. Yet, physical education is failing, one in five children achieve the daily physical activity recommendations and only a minority of adults continue activities experienced in the curriculum [26,27]. Despite the shortcomings of physical education, it should be treated as the primary means to create effective change. Lander et al. [28] systematically reviewed physical education intervention effectiveness with 25 studies significantly improving FMS in both primary and secondary education. Introducing resistance training into the school curriculum has demonstrated value to improving movement competency and athletic performance of secondary school children [21]. Strength and conditioning based activities can provide an alternate avenue to the traditional physical education structure which may benefit pupils that favour movement orientated lessons [29].

To further enhance knowledge and understanding of strength and conditioning in physical education there is a need to simultaneously acknowledge changes to psychological constructs, a key component of physical literacy [10]. Previous research has consistently considered motivation as a contributing factor to motor learning [30,31]. Therefore, changes to motivation may coincide with learning new skills. One theory that maybe worthy of consideration is self-determination theory [32,33]. This is a macro-theory of human motivation, and provides a framework to understand motivation across a variety of domains, including sport and exercise settings [32,33]. Self-determination theory recognises the degrees to which the three basic psychological needs of autonomy, competence and relatedness contribute to the quality of motivation [32]. Self-determination theory identifies types of motivation and individuals goals, examining the influence of intrinsic motivation, extrinsic motivation or amotivation and the separate behavioural outcomes [32,34]. Self-determined motivation regulations have been attributed to more favourable adaptive outcomes in comparison to amotivation [35]. Thus, drawing on self-determination theory, the effectiveness of an intervention in physical education will rely on the extent to which youths psychological needs are met, and subsequently the extent to which they are intrinsically motivated to take part and improve. Strategies aiming to improve the strength of intrinsic motivation in youths should therefore be employed [31].

Owing to the large changes in self evaluations, unstable self-perceptions and fluctuating emotions often experienced by adolescents, considering the impact of an intervention on additional psychological constructs beyond motivation is also important [36,37]. In physical education, self-efficacy (i.e., individuals situational belief in their ability) is a significant precursor to physical activity levels and influences achievement [38,39,40]. Previous school-based interventions focusing on the mastery of skills, encouragement and moderately intense exercise to enhance self-efficacy have been fruitful [41]. For instance, Dishman et al. [41] revealed that manipulating self-efficacy resulted in increased physical activity levels among adolescent girls. However, resistance training alone has previously been unsuccessful at improving the self-efficacy of children [42]. As such, considering specific strategies to enhance children’s self-efficacy while delivering resistance training within physical education is pertinent. Self-esteem is another psychological construct related to participation in physical activity in adolescents [43]. Coopersmith [44] defined self-esteem as one’s beliefs in competence, success, and worthiness. Adolescence is a period during which individuals often experience large declines in self-esteem, only slowly recovering into adulthood [45]. Such declines often occur due to an increase in peer to peer comparison and as such, an emphasis on the development of self-esteem during adolescence to offset the potential negative consequences of such comparisons is important [46] Previous research conducted in physical education has shown an aerobic exercise intervention was ineffective at enhancing children’s self-esteem [47], but the development of movement skills and muscular strength to facilitate preparation for recreational activity in adulthood may be useful [46].

Strength and conditioning research has gathered momentum in the last few decades, providing evidence that it is both safe and can improve movement competency, performance, health and well-being measures in children and adolescents [18,19]. Despite the surge in strength and conditioning research over the last few decades, few studies have simultaneously explored the effects of strength and conditioning interventions on both physical and psychological constructs. Therefore, the aim of the study was to explore the feasibility of integrating strength and conditioning based activities into physical education. We also aimed to examine the effects of a strength and conditioning intervention on AMSC in secondary school boys and girls. In addition, we sought to determine the effects of an intervention on motivation to exercise, physical self-efficacy, and global self-esteem in secondary school children. It was hypothesised that strength and conditioning in physical education could effectively enhance AMSC of adolescent school children and positively impact upon psychological constructs.

## 2. Methods

### 2.1. Experimental Summary

Forty-six boys and girls aged 11–14 years were included in the study, twenty-three participated in the intervention, with twenty-three participants recruited from another school, pair-matched by sex and estimated maturity, and assigned to control groups; groups were in single-sex classes. Three entire physical education classes were selected at random by the physical education staff to form the intervention groups; two female classes and one male class to account for the typically larger male class sizes at the school. Participants remained in their respective physical education classes for the duration of the intervention. The summer school term afforded the study a nine-week period to complete pre and post intervention data collection and the delivery of the intervention. After accounting for data collection periods, a six-week period remained to deliver the intervention. The pupils at both schools received three physical education sessions every two weeks on a rolling basis. Therefore, three strength and conditioning sessions were delivered every two weeks, totaling nine sessions over the six-week time span. Intervention groups replaced all physical education with strength and conditioning sessions that focused on promoting AMSC, while the respective control groups participated in normal physical education class. The control groups received typical physical education lessons characterised by teaching games for understanding, sampling a variety of team and individually based sports; a teaching method popular for motor learning in physical education [48]. To assess the effect of the intervention, participants completed measures of movement competency, physical performance, motivation to exercise, physical self-efficacy, and global self-esteem before and after the intervention.

### 2.2. Participants

Boys and girls in years 7–9 in a secondary school in Wales were invited to participate in the study. Seventy participants (male = 31; female = 39) were initially selected from one secondary school to form the intervention groups. The school was located in an area of low socioeconomic status, with child poverty levels above the national average. With high levels of pupil absence during testing and intervention sessions, only 23 participants (male = 10, female = 13) completed the baseline and post intervention testing and attained the minimum attendance inclusion criteria of five out of the nine sessions. The inclusion criteria of five sessions was based on previous research that reported improvements in motor competency after only four weeks of training [49]. Twenty-three control participants from another school, also from an area of lower socio-economic status were pair-matched to the intervention group by sex and estimated somatic maturation [50]. Control participants were pair-matched by sex and estimated maturity to account for the varying adaptations in response to training stimuli in youth. Table 1 indicates the age, estimated maturity, and anthropometric data for all participants in each group. Parental consent and participant assent were obtained, and ethical approval was granted by the university research ethics committee prior to the onset of data collection.

### 2.3. Athletic Motor Skill Competencies

All tests were performed in a school sports hall during physical education lessons. Prior to completing any of the measures, at the start of the session, all participants completed a standardised five minute warm up. Athletic motor skill competencies were examined by both process and product orientated methods of assessment. For process orientated competency measures, the resistance training skills battery (RTSB) [51] and tuck jump assessment (TJA) [52] were selected with demonstrated reliability in adolescent populations [51,53]. The full RTSB consists of six skills, however for the purpose of the study a modified version was used which consisted of the squat, lunge, front support with chest touch and push-up. The suspended row and standing overhead press were removed due to time constraints of the school environment and the intervention including limited pulling or overhead pushing movements.

Prior to performing the RTSB, participants were given a standardised demonstration by the primary researcher, detailing the start and end positions of the movements. Participants were then given an opportunity to perform a practice set of each movement beforehand but received no cueing or feedback while performing the RTSB. Each individual exercise of the RTSB was performed twice for four repetitions and graded out of four or five performance criteria dependent on the exercise. Observing the performance criteria of each movement presented in Table 2, the assessor selected the best repetition from each set and awarded a point for each performance criteria performed correctly by the participant. The assessors’ score of each exercise totals the two sets performed of bilateral movements (squat, front support and chest touch, press up) and one set per limb for the unilateral lunge exercise. All points attained for each movement were totalled together to form a score indicating a participant’s level of AMSC. The highest total score that could be attained was 38 comprised of a maximum ten points from bodyweight squat, lunge, front support with chest touches and eight points for push-up. Smith et al. [54] assessed AMSC in a school-based adolescent population using the RTSB classifying competent to be attaining all performance criteria in a movement. Near competence was reported as achieving all but one criterion and less than near competent not achieving two or more performance criteria. The authors revealed the prevalence of competence to be between 3.3–27.9% for the modified RTSB in this study. [54]. Furthermore, less than near competent prevalence ranged from 33.4–83.2% [54]. With regard to previous research, a higher score was indicative of high levels of competency. Participants were filmed and performance rated retrospectively by the lead researcher using video footage. Two-dimensional (2D) cameras (iPad mini 2, Apple, Cupertino, CA, USA) were positioned in the frontal plane at a height of 0.70 m and a triangulated distance of 5 m from the centre of the capture area.

Competency in jumping and landing mechanics was also assessed using the TJA [52]. Participants were instructed to jump as high as possible aiming to spend little time on the ground to minimise ground contact time and to pull their knees up to their chest on each jump. Participants were instructed to jump and land on the same spot for a period of 10 s. Two-dimensional (2D) cameras (iPad mini 2, Apple) were positioned in the frontal plane and sagittal plane at a height of 0.70 m and a distance of 5 m from the capture area. Participants performance was rated retrospectively by the lead researcher using video footage observing the criteria given in Table 3. The TJA scores participants on 10 movement criteria during repeated tuck jumps completed in place for 10 s. The TJA uses a negative scoring system out of 10 marks, with lower scores representing higher levels of jumping, landing, and rebounding competency.

A product orientated measurement was used to assess lower limb strength via the standing long jump assessment. The standing long jump was recorded using a tape measure with all participants having a practice jump and three recorded jumps. The acceptable reliability (ICC = 0.94) has previously been published for the long jump in a youth population [55].

### 2.4. Intra-Rater Reliability of Competency Measures

Aligned to previous studies utilising the RTSB and TJA [21,53] the intra-rater reliability of the primary researcher on the project was determined. Ten participants pre-intervention data collection videos of the RTSB and TJA were selected at random to be included in the intra-rater reliability measures. The primary researcher graded videos (not counting towards data in the study) prior to conducting the reliability study to become familiarised to the RTSB, TJA, and the respective grading criteria. Videos were graded on three separate occasions with over two weeks in-between each set of gradings. The primary researcher kept all scores from the three gradings separate to ensure no cross checking occurred. The two-week break between gradings ensured the rater did not score participants based on memory of the previous grading.

### 2.5. Psychological Measures

Questionnaires were input on to an online software (quicktapsurvey.com) and downloaded onto iPads (iPad mini 2, Apple) that allowed responses to be collected while maintaining the structure and scale of the questionnaires.

Motivation to exercise was assessed using the Behavioural Regulation in Exercise Questionnaire-2 [56] which provides a relative autonomy index, with a higher relative autonomy index being attributed to higher levels of motivation to exercise. The Behavioural Regulation in Exercise Questionnaire-2 comprising 19 questions, framed as the following example “I exercise because other people say I should” applying a five-point Likert scale ranging from “Not true for me” to “Very true for me” (0-4). All five subscales of motivation (intrinsic, identified, introjected, external, and amotivation) have been demonstrated to have acceptable reliability and consistency (Cronbach alpha > 0.7) [56].

Physical self-efficacy was assessed using the six-item Perceived Physical Ability Scale for Children due to its good levels of reliability (Cronbach alpha = 0.72) [57]. The scale provides four statements per item, for example, participants would select one of the following: “I run very slowly”, “I run slowly”, “I run fast” or “I run really fast”.

Participants’ global self-esteem was examined using the Rosenberg Self-Esteem Scale [58]. The 10-item scale uses a four-point Likert scale scored 3-0 ranging from “strongly agree” (3) to “strongly disagree” (0) in answer to the following example: “On the whole, I am satisfied with myself”. For five questions scoring is reversed, for example: “At times, I think I am no good at all” in this case “strongly agree” would score 0 and “strongly disagree” 3.

Participants were also asked to indicate their physical education enjoyment levels, ranging from “yes” to “sometimes” to “no”. Participants in the intervention groups were also asked if they enjoyed the intervention over traditional physical education lessons with the option to answer either “yes” or “no” during post-intervention data collection.

### 2.6. Intervention

The intervention was designed and delivered by the lead researcher, who had years of experience delivering strength and conditioning in schools. The exercises included in the intervention were selected on the basis of targeting improvements in AMSC [17]. To stimulate strength adaptations, resistance was provided using body weight, resistance bands, or medicine balls. The basic resistance training equipment were incorporated into games, challenges or short periods of teaching to learn technique. Games and challenges utilised an individualised, constraints-led approach, by manipulating task and environmental constraints to promote the development of the AMSC [59]. Many exercises were integrated into games to allow for exposure to AMSC to be achieved in an enjoyable and engaging manner for the pupils. The intervention considered total exposure to movements as the primary measure; providing the coach with a novel and flexible approach to set and repetition prescription. The intervention was designed to allow individual differentiation between participants, where pupils could be progressed or regressed on the same movements when deemed appropriate by the coach (i.e., the primary researcher). Table 4 provides an example training session to demonstrate how the constraints-led approach was used to incorporate AMSC whilst considering psychological principles of self-determination theory.

The intervention was grounded on key theoretical principles from self-determination theory [33] and achievement goal theory [60]; aiming to maximise participants’ motivation to exercise during the intervention and their overall enjoyment. The primary researcher reflected with the physical education staff throughout the intervention with the intention of evolving the intervention as it developed, aiming to enhance pupil enjoyment.

### 2.7. Statistical Analysis

Statistical analysis were performed using IBM SPSS statistics version 24 (IBM corp, Armonk, NY, USA). The intraclass correlation coefficient (ICC) calculated reliability of the TJA, total of the RTSB, and the four RTSB exercises individually. Kappa (k) was calculated for each individual performance criteria of each skill in the RTSB and the TJA. Additionally, the typical error of the modified RTSB and TJA were calculated on the reliability sample. Post-intervention gains in competency greater than the typical error were considered to be meaningful and gains more than double the typical error were considered to be large. Shapiro–Wilk tests for normality were used to determine the normality of data distribution for each of the measures. Parametric and non-parametric statistical analyses are presented as mean ± standard deviation and as median and interquartile ranges, respectively. To assess the differences between both males and females and intervention and control groups, independent T-tests (parametric) and Mann–Whitney U tests (non-parametric) were used to identify any statistically significant differences between the sub-groups using data collected at baseline. Paired T-tests (parametric) and Wilcoxon signed rank tests (non-parametric) were used to assess differences between each measure from to pre- to post-intervention within each group. In addition to T-test’s, effect sizes were calculated in Microsoft Excel (Version 16.30) using Cohen’s d statistic [61]. To identify statistically significant differences from pre- to post-intervention between-group comparisons, independent T-tests and Mann–Whitney U tests were used on the change scores from pre-to-post intervention. For all statistical analysis alpha *p* > 0.05 was considered to be significant. Each movement of the RTSB was expressed as the median + first (Q1) and third (Q3) quartile range and minimum and maximum values.

## 3. Results

### 3.1. Intra-Rater Reliability

The intraclass correlation coefficient indicated excellent reliability for RTSB sum score (ICC = 0.97) and good to excellent reliability for the four selected individual exercises (ICC = 0.88–0.94). Typical errors for the sum of the four selected exercises was 1.65 with a typical error of the individual exercise scores ranging from 0.48–0.80. For the individual rating criteria of the RTSB, the average kappa (K) statistic revealed fair to moderate agreement (K = 0.357–0.609). Intra-class correlations also demonstrated that the TJA had excellent reliability (ICC = 0.91), with individual criteria revealing poor to almost perfect strength of agreement (K = 0.286–1.000).

### 3.2. Lower Limb strength and AMSC

The standing long jump was the only variable found to be normally distributed. When combining data for control and intervention groups at baseline, males displayed significantly greater standing long jump (142 ± 23 vs. 128 ± 21; *p* > 0.05) compared to females. The total score for the modified RTSB (22 (20–24.5) vs. 22.5 (20.25–25.75)), TJA (5.5 (3–7) vs. 4.5 (4–6)) did not show any significant differences between males and females at baseline.

The scores for all measures pre- and post-intervention are shown in Table 5, which indicate within group changes and statistical significance from baseline to post intervention for non-normally distrusted measures. Both male and female intervention groups experienced a significant increase in total RTSB score (*p* < 0.05). The magnitude of the change in median total RTSB was greater than the typical error for the boys (+4) but not the girls (+1); however, the shift in the Q1 and Q3 range was above the typical error in the female intervention group (22.0–28.0 vs. 24.0–30.0). In comparison, neither male or female control groups experienced a significant change in RTSB score post intervention, and the magnitude of change observed was lower than the typical error. When comparing the effects of the intervention on RTSB score, the male intervention group displayed a significant improvement compared to the control group (*p* > 0.01), yet there were no differences between the two female groups. Figure 1 shows the individual responses for combined male and female intervention groups, with data indicating 17 in 23 participants (74%) increased total RTSB score by more than the typical error (change in total score ≥ 2), with 12 in 23 (52%) increasing performance by more than double the typical error (change in total score ≥ 4). Contrastingly, only 4 in 23 (17%) participants in the control group recorded an improvement of 4 points or more.

For the individual exercises, the male intervention group experienced statistically significant improvements in the squat and lunge exercises (*p* > 0.05), and those changes were greater than the typical error associated with each movement (>0.8). However, the male control group also made a significant and meaningful change in their squat performance, with improvements above the level of typical error. The female intervention group only experienced a significant improvement in the front support brace and chest touch movement (*p* > 0.05), which was also greater than the typical error (Table 6). Inter group comparisons revealed no significant changes in RTSB.

At baseline, there were no significant differences between intervention and control groups except for standing long jump (*p* = 0.010). There were no significant differences post intervention in the standing long jump for all four groups. Pre- to post-intervention, the effect sizes were trivial for the male intervention, male control and female control groups (d ≤ 0.11). The female intervention group displayed a small positive change (d = 0.42).

### 3.3. Motivation, Physical Self-Efficacy and Self-Esteem

Combined data for control and intervention groups at baseline displayed significantly greater global self-esteem (22.5 (20–27.5) vs. 20 (17–22); *p* > 0.05) in males compared to females. Relative autonomy index (11.17 (5.58–14.54) vs. 11.75 (7.04–13.93)) and physical self-efficacy (17.50 (15.00–21.25) vs. 17.50 (15.25–18.75) did not show any significant differences between males and females at baseline.

Few significant differences were observed in motivation, physical self-efficacy and global self-esteem. The male intervention group showed a significant within-group increase from pre- to post-intervention in relative autonomy index (motivation to exercise) (*p* > 0.05). There was not a significant difference between male intervention and control groups post-intervention for physical self-efficacy and global self-esteem. However, a significant within-group decrease in global self-esteem in the male control group (*p* > 0.05).

Prior to starting the intervention, 87% of the intervention participants and 74% of the control participants reported that they enjoyed physical education. At baseline, the highest level of reported enjoyment was in the male intervention and control groups both reporting 90%. Post-intervention, three-quarters of the intervention group (17 in 23) indicated that they enjoyed the intervention more than conventional physical education. The average attendance for the intervention considering pupils that attained the inclusion criteria for the study was 83%

## 4. Discussion

The aim of the study was to develop understanding to the feasibility of implementing strength and conditioning into physical education. In addition, to examine the effects of a strength and conditioning intervention on AMSC, motivation to exercise, physical self-efficacy and global self-esteem in school children. The results demonstrate that it is feasible to integrate strength and conditioning into physical education, particularly when considering the self-reported preference of the intervention compared to conventional physical education. However, while the intervention was effective in improving the competency of those who attended physical education classes which incorporated strength and conditioning, low attendance was a barrier to the reach of the program. Process-orientated measures of competency were the most sensitive to change. The intervention demonstrated the ability to improve young school children’s AMSC, as evidenced by the improvement in RTSB score. However, similar improvements were not experienced in TJA, which is another process-orientated measure that assesses jumping, landing and rebounding mechanics. One potential explanation surrounding the non-significant change in TJA is that it requires high levels of effort [52], and force production, requiring underlying levels of strength. Similarly, much like the TJA, the standing long jump mirrors the need to produce high levels of force and is used as a surrogate to assesses lower body muscular strength levels [62]. Yet, standing long jump performance did not improve in either the male or female intervention groups. Cumulatively, it may be that the intervention failed to improve the underlying strength qualities needed to enhance tuck jump and long jump performance. Longer interventions that progress to higher intensities of resistance training are likely needed to support further training gains. Signifying for improvements in strength to occur, strength and conditioning should be considered long-term within physical education. For example, in a recent meta-analysis by Lesinski et al. [63] the authors identified the dose-response of training interventions in youths that maximise muscular strength adaptations to be much longer in duration (>23 weeks). Furthermore, the authors reported much higher intensities (80–89% of one repetition maximum) than the current intervention to maximise muscular strength [63]. Although resistance-based exercises were utilised in the current study, the primary aim was to improve movement competency. As such, the total volume of exercise and the low resistance used may not have been enough to stimulate increases in gross muscular strength in just six weeks.

For shorter interventions, process measures should be the primary consideration with movement competency and fundamental movement skills being shown to positively adapt in as little as four weeks [49,64]. One potential explanation for the positive responses that children show to relatively short interventions is their heightened neuroplasticity. This means movement competency may be highly adaptable in children, with neuroplasticity strongly related to motor learning [65,66]. The intervention in the present study was successful at improving AMSC, evidenced by improvements in RTSB score. The results of the study identify the importance of exposing youth to a variety of athletic movements to help develop competency in those movements, which follows the principle of training specificity. Athletic motor skill competencies were embedded into games and challenges in physical education classes and constraints introduced to challenge task demands, with those competencies improving pre to post intervention. Developing movement competency is important to enable children to progress to more complex and demanding exercise, thus teaching movement skills in a periodised manner should take precedence in any athletic development program [20]. The practical implications of changes in AMSC occurring as a result of a short intervention of this nature indicate strength and conditioning embedded in the school physical education curriculum can positively influence motor skill competency. Rapid changes in AMSC reflect the neuroplasticity of youths and young children’s heightened ability to learn new skills [65,67]. Further, competency in the AMSC in youths are low [54], providing large scope to improve such skills.

The only changes pre- to post-intervention for physical self-efficacy and global self-esteem were observed in the male control group, who experienced a significant decrease in global self-esteem. However, the change was not significant when compared with the intervention group. Puberty is a volatile time for self-esteem levels, fluctuations in self-esteem during adolescence may occur as limitations in cognitive ability to create self-perceptions diminish [45,68]. Girls participating in free weight training have previously demonstrated an increase in self-perception after an eight-week intervention, while males’ self-perceptions remained constant [69]. The lack of change in physical self-efficacy shown in the current study could be attributed to the short nature of the intervention, not allowing for positive alterations in movement competency to translate to psychosocial changes. Despite intervention groups displaying improvements in AMSC, the physical changes did not translate to changes to physical perceptions. Physical self-efficacy endorses engagement, skill acquisition and learning [70], however changes in AMSC translating to increased physical self-efficacy likely take longer than the short duration of the intervention. Furthermore, the magnitude of adaptations to AMSC may need to be greater to positively influence psychological constructs.

The significant improvement in motivation to exercise experienced by the male intervention group maybe due to differences in needs satisfaction contributing to self-determined motivation between males and females. Ntoumanis [71] identified that physical education teachers support of students’ psychological needs was related to greater self-determined motivation. Significantly, between sex differences in adolescent pupils physical education needs satisfaction has shown relatedness to be more powerful in females than males [71]. Therefore, the intervention may have appealed more to males than females, given the focus of activities with many tasks focusing on self-improvement and competency. Another potential explanation for the improvement in motivation to exercise could be attributed to the high levels of physical education enjoyment reported by the male intervention group. The high levels of self-reported physical education enjoyment could have resulted in increased adherence to the strength and conditioning intervention and thus translated to enhanced motivation to exercise.

### Limitations

A confounding issue experienced in this demographic was the low attendance rate of pupils within the physical education lessons. Poor attendance rates reflect the realities of many schools, particularly those based in areas of low socio-economic status. Unauthorised absence can be up to four times higher in areas of lower socio-economic status in comparison to areas of higher status [72]. Despite the low levels of attendance, and even with some pupils only participating in five out of nine sessions, the intervention was still effective overall at improving AMSC. The intervention ran in the summer term, which also contributed to low attendance, with some pupils missing sessions due to field trips or being absent at the end of term when post-intervention testing was conducted. Despite the limitations, the findings provide evidence to support the feasibility of working in this demographic. Accordingly, the findings from this study provide foundational knowledge that future studies can build on. In studies with higher levels of attendance and longer durations, the intervention could be progressed further with increasing intensity while ensuring good movement quality. The study was only conducted on a small sample so findings should not be generalised to a wider population, however the findings provide initial evidence of the potential benefits that strength and conditioning interventions may have on improving AMSC in the school setting. The minimal change in psychological constructs may indicate that there was insufficient emphasis on these areas within the program or that positive outcomes take longer to manifest. Therefore, studies with larger sample sizes, longer intervention durations and greater considerations of psychological constructs may be warranted.

This study did not investigate the impact the primary researcher and the teachers had on the intervention and control groups respectively. Substantial evidence supports teachers influence on student outcomes in fundamental movement skills and physical activity [28]. Autonomy supportive environments stimulated by teachers enhances student motivation in physical education [73]. In addition, the primary researcher, with extensive experience of youth strength and conditioning coaching, was unbound by the restrictions of the physical education curriculum. Accordingly, alternate approaches, coaching style, knowledge and understanding of how to communicate with youths may have been exhibited in comparison to the teachers in the control groups. Future research could adopt approaches that have previously investigated motivational climates and the psychological needs satisfaction of students in physical education [73,74]. In addition, future strength and conditioning studies could adopt self-reported physical activity measures to understand the influence of students external behaviour, as featured in previous physical education intervention studies [75].

## 5. Conclusions

The findings from this study provide novel evidence of the efficacy of a short intervention to improve AMSC, while providing important insights to improve subsequent interventions. Strength and conditioning can feasibly be used as means to improve movement competency in children, as has previously been outlined by leading global physical activity guidelines [3]. This study provides an insight into the practicalities of delivering strength and conditioning in schools of lower socio-economic status. Future studies can build on the findings to enhance knowledge and understanding of how to implement resistance training in physical education. In particular, strategies to improve attendance in physical education interventions delivered in schools in areas of low socio-economic status could help expose more school-children to the benefits of strength and conditioning. Due to the known associations between movement competency, physical literacy, physical activity and health-related fitness, greater consideration to the use of strength and conditioning in physical education to enhance the health status of youths should begin. The findings from this study can inform future studies to examine the potential of resistance training to improve AMSC, physical literacy, and as an alternate means to enhance pupil physical education enjoyment.

## Figures and Tables

**Figure 1 sports-08-00138-f001:**
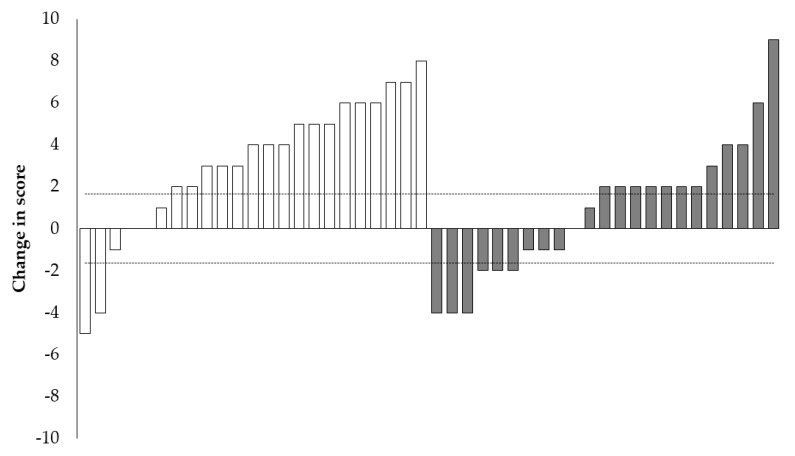
Individual participant changes in modified RTSB, intervention participants in white and control participants in grey. Dotted line shows the typical error (±1.65) calculated from the intra-rater reliability of the primary researcher.

**Table 1 sports-08-00138-t001:** Age, maturity and anthropometric data for all groups.

Group	Age (Year)	Maturity Offset (Year)	Height (cm)	Weight (kg)
Male intervention group (N = 10)	12.07 ± 0.30	−1.57 ± 0.74	151.40 ± 7.38	48.15 ± 12.54
Male control group (N = 10)	11.67 ± 0.35	−1.53 ± 0.71	150.27 ± 12.84	44.07 ± 14.21
Female intervention group (N = 13)	12.56 ± 0.97	0.20 ± 1.00	152.87 ± 7.96	46.35 ± 11.16
Female control group (N = 13)	11.90 ± 0.76	0.15 ± 0.91	150.10 ± 8.05	45.65 ± 11.29

Notes: Maturity offset = time in years pre-to-post peak height velocity.

**Table 2 sports-08-00138-t002:** Performance criteria of the movements from the modified Resistance Training Skill Battery (RTSB) [51].

AMSC		Performance Criteria
Body weight squat	1	Feet are shoulder width or slightly wider apart and facing forward
	2	Back is kept straight and stable throughout the movement
	3	Knees point in the same direction as feet during movement
	4	Heels remain on the floor throughout the movement
	5	Thighs are parallel to the floor at the bottom of the movement
Push-up	1	Hands are shoulder width or slightly wider apart
	2	Head, back and hips are held in a straight line throughout the movement
	3	Body is lowered until elbows are at a 90 degree angle
	4	Shoulders are held down and away from ears (shoulders are not shrugged)
Lunge	1	Takes an exaggerated step forward and lands heel first
	2	Torso is kept upright and stable at all times (no twisting)
	3	Knee of rear leg is almost touching the floor (approx. 10cm)
	4	There is alignment between hip, knee and foot of each leg
	5	Returns to starting position in one movement
Front Support with chest touches	1	Straight line through legs, hips, shoulders and head
	2	Feet are approximately shoulder width apart
	3	Minimal rotation of body while changing hand placement (approx. 10 cm is acceptable)
	4	Both feet remain on the ground throughout the entire trial
	5	Chest touches are performed in a controlled manor

Notes: AMSC = athletic motor skill competencies.

**Table 3 sports-08-00138-t003:** Assessment criteria for the Tuck Jump [52].

AMSC		Assessment Criteria
Tuck Jump	1	Lower extremity valgus at landing
	2	Thighs do not reach parallel (peak of jump)
	3	Thighs not equal side-to-side (during flight)
	4	Foot placement not shoulder width apart
	5	Foot placement not parallel (front to back)
	6	Foot contact timing not equal
	7	Excessive landing contact noise
	8	Pause between jumps
	9	Technique declines prior to 10 s
	10	Does not land in the same footprint
		(excessive in-flight motion)

Notes: AMSC = athletic motor skill competencies.

**Table 4 sports-08-00138-t004:** Example training session incorporating session structure, exercises, exposure, implementing constraints-led approach and psychological considerations.

					Constraint		Self-Determination		
Structure	Set up	Target AMSC	Exercises	Exposure	Type	Implementation	Type	Implementation	Additional Strategies
Reactive Cone Challenge	Self-selected pairs	Anti-rotation/Core Bracing	Brace with shoulder tap	10	Task	All exercises are included into a fast-paced game governed by the coach. The coach controls the speed, and the varying transitions between each exercise. The coach controls the task constraint.	Autonomy	For this game choice can provided to pupils via self-selection of partner to compete against.	
		Lower Body Bilateral	Squat	20					
		Jumping, Landing and Rebounding Mechanics	Pogo jump	60					
			Tuck jump	30					
			Drop landing	5					
			Hopping	30					
Learning to Lunge	In pairs	Lower Body Bilateral	Lunge Lunge isometric hold + Perturbations	10	Task	The constraints on the lunging task are imposed by the coach’s language, feedback and reflective questions	Relatedness	For this part of the session the coach selects the teams, there may be some initial rebellion, but feelings of relatedness will be stimulated post session.	
				30’s	Environment	The unpredictable perturbations caused by the partner while holding the lunge position causes reorganisation of degrees of freedom to maintain lunge position.			
Team Relay	Two teams	Lower Body Bilateral	Lunge isometric hold	30’s	Task & Environment	Holding the AMSC while in a fast-paced game will stimulate (re)organisation as pupils react the changing game and environment around them.			
Movement Assault Course	Pre-set up assault course including: Box tops, balance beams, foam balance pods, hurdles and bench	Lower Body Bilateral	Squat	12	Task & Environment	The assault course changes the perceptual variables towards AMSC, different surfaces, interactions, distances and targets changes the affordances of the AMSC. The unknown environment stimulates (re)organisation of the motor skills. There are also task-constraints imposed by the coach i.e., what, how and when to perform exercises on the assault course.	Competence & Autonomy	The coach should provide continual positive feedback. Provide pupils with reference points for their own competency and to encourage learning via positive feedback. The assault course should also have routes of varying difficulty, pupils are given free choice to decide which path to take.	Providing different routes of varying difficulty on the assault course while allowing pupils to decide the route they take helps provide an optimal challenge. Pupils will decide the route they want to take based off their perceived competency and motivation to engage in a more difficult task. It allows pupils to explore new exercises and challenge rather than being instructed to do so.
		Lower Body Unilateral	Lunges	12					
		Anti-rotation and Core Bracing	SL squat	18					
		Jumping, Landing and Rebounding Mechanics	SL balance	30’s					
			Crab walk	60m					
			Bear crawl	60m					
			CMJ to box	6					
			CMJ over hurdle linear	18					
			CMJ over hurdle lateral	18					
			SL lateral landing	18					

Notes: Exposure = total amount of repetitions/ seconds/ distance covered of an exercise in the specific task; CMJ = countermovement jump; SL = single leg; s = seconds; m = meters.

**Table 5 sports-08-00138-t005:** Participants Pre- and Post- intervention scores (Mdn and 1st–3rd Quartile) and significance.

	Male Intervention Group (N = 10)				Male Control Group (N = 10)				Female Intervention Group (N = 13)				Female Control Group (N = 13)			
	Pre		Post		Pre		Post		Pre		Post		Pre		Post	
	Mdn	IQR	Mdn	IQR	Mdn	IQR	Mdn	IQR	Mdn	IQR	Mdn	IQR	Mdn	IQR	Mdn	IQR
RTSB	22.00	(20.00–25.25)	26.00 *	(23.25–28.00)	21.00	(17.00–24.00)	21.00	(19.25–24.75)	25.00	(22.00–28.00)	26.00 *	(24.00–30.00)	22.00	(19.00–24.00)	21.00	(20.00–26.00)
TJ	3.50	(2.25–6.75)	4.00	(3.25–5.00)	7.00	(5.00–7.00)	5.50	(4.00–6.75)	4.00	(4.00–6.00)	5.00	(4.00–6.00)	5.00	(4.00–6.00)	4.00	(4.00–5.00)
RAI	9.00	(5.60–12.65)	12.88 *	(10.25–14.83)	12.71	(6.44–14.63)	11.38	(7.44–14.63)	11.40	(5.80–12.80)	12.58	(11.00–15.25)	12.65	(8.67–14.00)	11.50	(5.25–13.75)
GSE	22.00	(19.25–23.00)	22.00	(20.00–27.25)	23.50	(20.25–28.50)	21.50*	(19.5–24.25)	20.00	(17.00–25.00)	21.00	(18.00–25.00)	20.00	(17.00–21.00)	20.00	(19.00–21.00)
PSE	18.50	(16.25–21.00)	19.00	(17.25–20.75)	17.00	(15.00–21.00)	18.00	(17.00–19.00)	18.00	(16.00–20.00)	18.00	(17.00–21.00)	17.00	(15.00–18.00)	17.00	(15.00–18.00)

Notes: * *p* > 0.05; Mdn = Median; IQR = Interquartile range; RTSB = Resistance training skill battery; TJ = Tuck jump; RAI = Relative autonomy index; GSE = Global self-esteem; PSE = Physical self-efficacy.

**Table 6 sports-08-00138-t006:** Participants Pre - and Post - intervention individuals exercises of RTSB scores Mdn (1st-3rd IQR) and significance.

	Male Experimental Group (N = 10)				Male Control Group (N = 10)				Female Experimental Group (N = 13)				Female Control Group (N = 13)			
	Pre		Post		Pre		Post		Pre		Post		Pre		Post	
	Mdn	IQR	Mdn	IQR	Mdn	IQR	Mdn	IQR	Mdn	IQR	Mdn	IQR	Mdn	IQR	Mdn	IQR
Squat	7.50	(6.00–8.00)	8.50 *	(8.00–9.75)	6.00	(4.00–8.00)	7.00 *	(6.00–8.00)	8.00	(6.00–8.00)	8.00	(8.00–9.00)	6.00	(6.00–8.00)	6.00	(6.00–8.00)
Lunge	4.00	(3.00–4.75)	5.50 *	(4.00–6.75)	5.00	(4.00–6.75)	4.00	(4.00–4.75)	7.00	(6.00–9.00)	8.00	(8.00–8.00)	6.00	(4.00–6.00)	6.00	(4.00–6.00)
FSB + CT	6.00	(6.00–6.00)	6.00	(6.00–6.00)	6.00	(6.00–6.75)	6.00	(6.00–6.75)	6.00	(6.00–6.00)	7.00 *	(6.00–8.00)	6.00	(6.00–8.00)	6.00	(6.00–8.00)
Press up	5.00	(3.25–6.00)	5.50	(4.25–7.50)	4.00	(2.00–4.00)	4.00	(4.00–4.00)	4.00	(4.00–5.00)	4.00	(3.00–4.00)	2.00	(2.00–4.00)	2.00	(2.00–4.00)

Notes: * *p* > 0.05; Mdn = Median; IQR = Interquartile range; RTSB = Resistance training skill battery; FSB + CT = Front support brace + chest touch.

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
