# Peer review of "The Effects of Strength and Conditioning in Physical Education on Athletic Motor Skill Competencies and Psychological Attributes of Secondary School Children: A Pilot Study"

_sports, 2020, doi:10.3390/sports8100138_

Round 1

Reviewer 1 Report

The study proposed is very interesting and is well structured, in the introductory part some words appear in yellow, and it is where population groups (adolescent, pediatric, ...) are not understood as a mixture, this question is very important to keep in mind since the sample is from 11 to 14 years old. It would be interesting to clarify these concepts.
At a methodological level, results, discussion and conclusions, the paper is very good.
I wish to congratulate the authors for the excellent work.

Reviewer 2 Report

Introduction

Very well written, with just few typos. In my opinion the authors could develop a bit more on the variables of self-esteem and self-efficacy, as they are dispatched in just 2-3 lines with no detail on previous studies analyzing them in PE/PA or so. I feel it would be good to give a bit more background on them; it would also help balancing the introduction a bit more, considering that motivation and the other dependent variables are treated with much more detail.

Methods

Experimental summary

Could the authors give a better insight on how the groups were created? Not sure whether they selected students regardless of the class-group they belonged, or (more common in such studies) they selected entire class-groups. If they selected students individually, how did they solve the problem of each of them participating in a different PE lesson?

How many times was PE carried out in the normal way? As far as I understand, EG kids had some regular PE classes and some intervention sessions…if that is the case, it is important to know what happened during regular PE. Also, were PE/sessions alternated within the schedule of the intervention, or how was that structured?

It is essential to report CG participants’ PE classes (what contents did they work on during the intervention period? What did they do?)

Please provide with information on PE frequency and duration in Wales

What did kids do during those regular PE classes?

Athletic Motor Skill competency

Although the authors refer to a previous work for detailed information on the tests RTSB and TJA, I feel readers would appreciate some more information:

  • for instance, how is the total score obtained (I mean, how are points assigned? Perhaps some example would be great; what is the role of the person observing the videos and rating them? How does his/her rate influence the final score?)?
  • Are there standard thresholds (authors say “the higher, the more competent”. That is fine; yet, are there threshold indicating poor, average, high competency?)?
  • How is the score of 38 reached (I mean, how does the rater assign points? I understand he/she observes the criteria in table 2, but then how does he/she establish a score per each? Is there a minimum/maximum?)?
  • Are there works using this tool and confirming it is reliable with only one rater? Usually, a one-rater evaluation is not considered very good, as it might be affected by biases (in this case for sure, considering the authors say it was the “lead researcher” who rated the movements, so well aware of the purpose and hypotheses of the study), and by subjective perceptions. It is quite common in these cases to have two or more raters, who would perform some pre-intervention training and establish a good (or higher) inter-rater reliability.

I work a lot with kids in schools as well, but I do not know this specific approach. I am a potential reader, so yes, I could go back and find the cited work to know everything I need; still, more detailed would be appreciated.

Intra-rater reliability of competency measures

I believe I understand that the only rater rated several times the same video and reliability was assessed based on “congruence” (the ratio at which the evaluations matched)?

I think this section should be rewritten towards increasing its clarity.

I know it is now impossible to go back and change this, but I still believe using more raters (and potentially unaware of the study purpose, or at least unaware of the group of the analyzed kids -experimental/control-) would increase the strength of the study. Again, this is something that cannot be changed; however, if the authors could bring a reference or two of studies using one rater and intra-rater reliability for these tests, it would be great.

Discussion

Most of the first paragraph should be erased: we already know the results, they are presented in the previous section and there is no need to repeat them in full. A brief reminder of the aim of the study as the entry line is fine, an entire paragraph on results is not. Also, the last sentence of the same first paragraph (from line 360) is more a summary/conclusion of the work than discussion. Authors should focus on trying to explain their results, compare them with previous literature, etc.

Limitations

I think the authors should also keep in mind and comment (here or somewhere else) some other limitations. For instance, have they assessed the out-of-school activity of the kids? Anything could contribute to increasing certain fitness parameters at this age, including participation in afternoon sports activities. Another important external factor could have been the teachers themselves, especially in psychological factors. EG was “trained” by the lead researcher (whose approach to the sessions, character, knowledge, ability to communicate, etc.) may have influenced the way kids perceived and absorbed the training. At the same time, EG also received some PE classes, therefore, the characteristics of the PE teacher/s also may have played a role. CG were from a different school, so they had a different teacher, with different ways to approach PE…in such studies, controlling these types of variables is very difficult, yet they should be mentioned either in “limitations” or where the authors consider it best.

Reviewer 3 Report

  • A brief summary (one short paragraph) outlining the aim of the paper and its main contributions.

This is a pilot study with the aim to identify the effects of a strength and conditioning intervention on athletic movement skill competencies in secondary school boys and girls. Secondly, determine the effects of an intervention on motivation to exercise, physical self-efficacy and global self-esteem in secondary school children. The introduction gives a clear idea of research and theory underpinning the study. The method are mostly well presented. I think the value of this study is mostly of what knowledge this study gives to feasibility in designing this kind of studies. The statistic result of such a small study is not so relevant per se.

  • Broad comments highlighting areas of strength and weakness. These comments should be specific enough for authors to be able to respond.

This is a well designed pilot study, which gives a lot of knowledge in the feasibility area of performing and evaluating this kind of intervention studies. However the authors do not recognize this aspects so much, instead they focus on the statistics of the results. The study is really small and it is really hard to draw any conclusions from, for example, 10 boys in the interventions group. I would like the authors to discuss more about “lessons learned” from the uncertainties of doing this kind of study. What can we learn from this study when designing a bigger study like it. What is the pit falls needed to be concerned?

  • Specific comments referring to line numbers, tables or figures. Reviewers need not comment on formatting issues that do not obscure the meaning of the paper, as these will be addressed by editors.

In the abstract section you mention that 46 school children were recruited but in the Method section you say that 70 were recruited in the intervention group, please clarify this.

In the method section please clarify how the inclusion of the control group was made. As I understand it you matched the control after the interventiongroup finished, i.e. how did you choose the 23 participants among the 70 in the control group. I understand that you chose age, maturity, height and weight, but I do not understand the argumentation behind this choice. Attending physical activity lessons could be more valid. Please discuss this in relation to the result.

 I am interested in seeing a table over the intervention group respective control group participation in the classes, was this also matched in the inclusion of controls? As you discuss this as an important aspect, it is important to show this data. Please also discuss the weakness of the study having only 23 of 70 finishing the study = 33%.

The intervention group received activities that was similar to the AMSC tests, discuss how this could affect the result of the study.

Last in the result you report that 87% of the participants in the interventions group enjoyed physical education before the intervention. What was the result of the control group regarding this aspect. This result have impact on the total result of the study, regarding what group of participants that this intervention reached. The result show that boys intervention group showed most effect, could this be due to they already were interested in doing exercises and was therefore more motivated to do this kind of exercises outside the intervention?

In the discussion part you discuss change regarding to weeks of attending this intervention. However as some of them attended only 5 sessions I do not know if attending X weeks this is the most relevant way of discussing it.

I think the authors should discuss much more in the limitation part. There are many aspects to highlight here (for example, see the comments above) and doing so do not diminish the value of the study, I think it is the opposite.

I do not understand the sentence on line 440 in conclusion “Considering the associations… “ please rewrite it.

I think that the last sentence in the conclusion it drawn to far considering the weakness of this study. I would like the authors to consider rewriting the whole conclusion part.

Round 2

Reviewer 2 Report

Thank you for your prompt response and congratulation on your effort
